# Artificial Intelligence (AI)-Empowered Echocardiography Interpretation: A State-of-the-Art Review

**DOI:** 10.3390/jcm10071391

**Published:** 2021-03-30

**Authors:** Zeynettin Akkus, Yousof H. Aly, Itzhak Z. Attia, Francisco Lopez-Jimenez, Adelaide M. Arruda-Olson, Patricia A. Pellikka, Sorin V. Pislaru, Garvan C. Kane, Paul A. Friedman, Jae K. Oh

**Affiliations:** Department of Cardiovascular Medicine, Mayo Clinic, Rochester, MN 55905, USA; Aly.Yousof@mayo.edu (Y.H.A.); attia.itzhak@mayo.edu (I.Z.A.); lopez@mayo.edu (F.L.-J.); ArrudaOlson.Adelaide@mayo.edu (A.M.A.-O.); Pellikka.Patricia@mayo.edu (P.A.P.); Pislaru.Sorin@mayo.edu (S.V.P.); Kane.Garvan@mayo.edu (G.C.K.); Friedman.Paul@mayo.edu (P.A.F.); Oh.Jae@mayo.edu (J.K.O.)

**Keywords:** cardiac ultrasound, echocardiography, artificial intelligence, portable ultrasound

## Abstract

Echocardiography (Echo), a widely available, noninvasive, and portable bedside imaging tool, is the most frequently used imaging modality in assessing cardiac anatomy and function in clinical practice. On the other hand, its operator dependability introduces variability in image acquisition, measurements, and interpretation. To reduce these variabilities, there is an increasing demand for an operator- and interpreter-independent Echo system empowered with artificial intelligence (AI), which has been incorporated into diverse areas of clinical medicine. Recent advances in AI applications in computer vision have enabled us to identify conceptual and complex imaging features with the self-learning ability of AI models and efficient parallel computing power. This has resulted in vast opportunities such as providing AI models that are robust to variations with generalizability for instantaneous image quality control, aiding in the acquisition of optimal images and diagnosis of complex diseases, and improving the clinical workflow of cardiac ultrasound. In this review, we provide a state-of-the art overview of AI-empowered Echo applications in cardiology and future trends for AI-powered Echo technology that standardize measurements, aid physicians in diagnosing cardiac diseases, optimize Echo workflow in clinics, and ultimately, reduce healthcare costs.

## 1. Introduction

Echocardiography (Echo), also known as cardiac ultrasound (CUS), is currently the most widely used noninvasive imaging modality for assessing patients with various cardiovascular disorders. It plays a vital role in evaluation of patients with symptoms of heart disease by identifying structural as well as functional abnormalities and assessing intracardiac hemodynamics. However, accurate echo measurements can be hampered by variability between interpreters, patients, and operators and image quality. Therefore, there is a clinical need for standardized methods of echo measurements and interpretation to reduce these variabilities. Artificial-intelligence-empowered echo (AI-Echo) can potentially reduce inter-interpreter variability and indeterminate assessment and improve the detection of unique conditions as well as the management of various cardiac disorders.

In this state-of-the-art review, we will provide a brief background on transthoracic echocardiography (TTE) and artificial intelligence (AI) followed by a summary of the advances in echo interpretation using deep learning (DL) with its self-learning ability. Since DL approaches have shown superior performance compared to machine-learning (ML) approaches based on hand-crafted features, we focus on DL progress in this review and refer the readers to other reviews [1,2] for ML approaches used to interpret echo. The AI advances could potentially allow objective evaluation of echocardiography, improving clinical workflow, and reducing healthcare costs. Subsequently, we will present currently available AI-Echo applications, delve into challenges of current AI applications using DL, and share our view on future trends in AI-Echo.

### 1.1. Transthoracic Echocardiogram

Transthoracic echocardiogram transmits and receives sound waves with frequencies higher than human hearing using an ultrasound transducer. It generates ultrasound waves and transmits to the tissue and listens to receive the reflected sound wave (echo). The reflected echo signal is recorded to construct an image of the interrogated region. The sound waves travel through soft tissue medium with a speed of approximately 1540 m/s. The time of flight between the transmitted and received sound waves is used to locate objects and construct an image of the probed area. The recorded echo data can be either a single still image or a movie/cine clip over multiple cardiac cycles. CUS has several advantages compared to cardiac magnetic resonance, cardiac computed tomography, and cardiac positron emission tomography imaging modalities. CUS does not use ionizing radiation, is less expensive, portable for point-of-care (POCUS) applications, and provides actual real-time imaging. It can be carried to a patient’s bedside for examining patients and monitoring changes over time. Disadvantages of TTE include its dependence on operator and interpreter skill, with variability in data acquisition and interpretation. In addition to operator variability, it includes patient specific variability (e.g., signal-to-noise ratio and limited acoustic window due to anatomical or body mass differences) and machine specific variability (e.g., electronic noise and post-processing filters applied to acquired images). Image quality plays an important factor for accurate measurements. Suboptimal image quality can affect all measurements and can result in misdiagnosis.

Diverse image types are formed by using cardiac ultrasound (Figure 1). The most common types used in clinics are:

B-mode: It is also called brightness mode (B-mode), which is the most well-known US image type. An ultrasound beam is scanned across the tissue to construct a 2D cross section image of the tissue.

M-mode: Motion mode (M-mode) is used to examine motion over time. For example, it provides a single scan line of the heart, and all of the reflectors along this line are shown along the time axis to measure temporal resolution of the cardiac structures.

Doppler ultrasound: A change in the frequency of a wave occurs when the source and observer are moving relative to each other, this is called the Doppler effect. An US wave is transmitted with a specific frequency through an ultrasound probe (the observer). The US waves that are reflected from moving objects (e.g., red blood cells in vessels) return to the probe with a frequency shift. This frequency shift is used to estimate the velocity of the moving object. In blood flow, the velocity of red blood cells moving towards and away from the probe is recorded to construct Doppler signals. The velocity of information overlaid on top of a B-mode anatomical image to show color Doppler images of blood flow.

Contrast enhanced ultrasound (CEUS): CEUS is a functional imaging that suppresses anatomical details but visualizes blood pool information. It exploits the non-linear response of ultrasound contrast agents (lipid coated gas bubbles). Generally, two consecutive US signals are propagated through the same medium, and their echo response is subtracted to obtain contrast signal. Since the tissue generates linear echo response, the subtraction cancels out the tissue signal, and only the difference signal from non-linear responses of bubbles remains. This imaging technique is used to enhance cardiac chamber cavities when B-mode US provides poor quality images. It is useful to detect perfusion abnormalities in tissues and enhance the visibility of tissue boundaries.

Strain imaging: This technique detects myocardial deformation patterns such as longitudinal, radial, and circumferential deformations, and early functional abnormalities before they become noticeable as regional wall motion abnormalities or reduced ejection fraction on B-mode cardiac images.

### 1.2. Artificial Intelligence

Artificial intelligence (AI) is considered to be a computer-based system that can observe an environment and takes actions to maximize the success of achieving its goals. Some examples include a system that has the ability of sensing, reasoning, engaging, and learning, are computer vision for understanding digital images, natural language processing for interaction between human and computer languages, voice recognition for detection and translation of spoken languages, robotics and motion, planning and organization, and knowledge capture. ML is a subsection of AI that covers the ability of a system to learn about data using supervised or unsupervised statistical and ML methods such as regression, support vector machines, decision trees, and neural networks. Deep learning (DL), which is a subclass of ML, learns a sequential chain of pivotal features from input data that maximizes the success of the learning process with its self-learning ability. This is different from statistical ML algorithms that require handcrafted feature selection [3] (Figure 2).

Artificial neural networks (ANN) are the first DL network design where all nodes are fully connected to each other. It mimics biological neurons for creating representation from an input signal, including many consecutive layers that learn a hierarchy of features from an input signal. ANN and the advance in graphics processing units (GPU) processing power have enabled the development of deep and complex DL models with simultaneous multitasking at the same time. DL models can be trained with thousands or millions of samples to gain robustness to variations in data. The representation power of DL models is massive and can create representation for any given variation of a signal. Recent accomplishments of DL, especially in image classification and segmentation applications, made it very popular in the data science community. Traditional ML methods use hand-crafted features extracted from data and process them in decomposable pipelines. This makes them more comprehensible as each component is explainable. On the other hand, they tend to be less generalizable and robust to variations in data. With DL models, we give up interpretability in exchange for obtaining robustness and greater generalization ability, while generating complex and abstract features.

State-of-the-art DL models have been developed for a variety of tasks such as object detection and segmentation in computer vision, voice recognition, and genotype/phenotype prediction. There are different types of models that include convolutional neural networks (CNNs), deep Boltzmann machines, stacked auto-encoders [4], and deep belief neural networks [5]. The most commonly used DL method for processing images that are CNNs as fully connected ANN is computationally heavy for 2D/3D images and requires extensive GPU memory. CNNs share weights across each feature map or convolutional layers to mitigate this. CNN approaches have gained enormous awareness, achieving impressive results in the ImageNet [6,7,8] competition in 2012 [8], which includes natural photographic images. They were utilized to classify a dataset of around a million images that comprise a thousand diverse classes, achieving half the error rates of the most popular traditional ML approaches [8]. CNNs have been widely utilized for medical image classification and segmentation tasks with great success [3,9,10,11,12]. Since DL algorithms outperform ML algorithms in general and exploit the GPU processing power, it allows real-time processing of US images. We will only focus on DL applications of AI-powered US cardiology in this review.

To assess the performance of ML models, data are generally split into training, validation, and test sets. The training set is used for learning about the data. The validation set is employed to establish the reliability of learning results, and the test set is used to assess the generalizability of a trained model on the data that are never seen by the model. When training samples are limited, k-fold cross validation approaches (e.g., leave-one-out, five-fold, or ten-fold cross validation) are utilized. In cross-validation, the data are divided randomly into k equal sized pieces. One piece is reserved for assessing the performance of a model, and the remaining pieces (k-1) are utilized for training models. The training process is typically performed in a supervised way, which involves ground truth labels for each input data and minimizes a loss function over training samples iteratively, as shown in Figure 3. Supervised learning is the most common training approach for ML, but it requires a laborious ground truth label generation. In medical imaging, ground truth labels are generally obtained from clinical notes for diagnosis or quantification. Furthermore, manual outlining of structures by experts are used to train ML models for segmentation tasks.

## 2. Methods and Results: Automated Echo Interpretation

We performed a thorough analysis of the literature using Google Scholar and PubMed search engines. We included peer-reviewed journal publications and conference proceedings in this field (IEEE Transactions on Medical Imaging, IEEE Journal of Biomedical and Health Informatics, Circulation, Nature, and conference proceedings from SPIE, the Medical Image Computing and Computer Assisted Intervention Society, the Institute of Electrical and Electronics Engineers, and others) that describe the application of DL to cardiac ultrasound images before 15 January 2021. We included a total of 14 journal papers and three conference proceedings that are relevant to the scope of this review (see Figure 4 for the detailed flowchart for the identification, screening, eligibility, and inclusion). We divided reports into three groups on the basis of the task performed: view identification and quality control, image segmentation and quantification, and disease diagnosis.

Current Echo-AI applications require several successive processing steps such as view labelling and quality control, segmentation of cardiac structures, echo measurements, and disease diagnosis (Figure 5). AI-Echo can be used for low-cost, serial, and automated evaluation of cardiac structures and function by experts and non-experts in cardiology, primary care, and emergency clinics. This would also allow triaging incoming patients with chest pain in an emergency department by providing preliminary diagnosis and longitudinally monitoring patients with cardiovascular risk factors in a personalized manner.

With the advancing ultrasound technology, the current clinical cart-based ultrasound systems could be replaced with portable point-of-care ultrasound (POCUS) systems or could be used together. GE Vscan, Butterfly IQ, and Philips Lumify are popular POCUS devices. A single Butterfly IQ probe contains 9000 micro-machined semiconductor sensors and emulates linear, phased, and curved array probes. While the Butterfly IQ probe using ultrasound-on-chip technology could be used for imaging the whole body, Philips Lumify provides different probes for each organ (e.g., s4-1 phased array probe for cardiac applications). GE Vscan comes with two transducers placed in one probe and can be used for scanning deep and superficial structures. Using POCUS devices powered with cloud-based AI-Echo interpretation at point of care locations could significantly reduce the US cost and increase the utility of AI-Echo by non-experts in primary and emergency departments (see Figure 6). A number of promising studies using DL approaches have been published for classification of standard echo views (e.g., apical and parasternal views), segmentation of heart structures (e.g., ventricle, atrium, septum, myocardium, and pericardium), and prediction of cardiac diseases (e.g., heart failure, hypertrophic cardiomyopathy, cardiac amyloidosis, and pulmonary hypertension) in recent years [13,14,15,16]. In addition, several companies such as TOMTEC IMAGING SYSTEMS GMBH, Munich, Germany and Ultromics, Oxford, United Kingdom have already obtained premarket FDA clearance on auto ejection fraction (EF) and echo strain packages using artificial intelligence. The list of companies and their provided AI tools is shown in Table 1.

### 2.1. View Identification and Quality Control

A typical TTE study includes the acquisition of multiple cine clips of the heart’s chambers from five standardized windows that are left parasternal window (i.e., parasternal long and short axis views), apical window (i.e., two, three, four, five chamber views), subcostal window (i.e., four chamber view, long axis inferior vena cava view), and suprasternal notch window (i.e., aortic arch view), right parasternal window (i.e., ascending aorta view). In addition to these, the study includes several other cine clips of color Doppler, strain imaging, and 3D ultrasound and still images of valves, walls, and the blood vessels (e.g., aorta and pulmonary veins). View identification and quality control are essential prerequisite steps for a fully automated echo interpretation.

Zhang et al. [16,17] presented a fully automated echo interpretation pipeline that includes 23 view classifications. They trained a 13-layer CNN model with 7168 labelled cine clips and used five-fold cross validation to assess the performance of their model. In evaluation, they selected 10 random frames per clip and averaged the resulting probabilities. The overall accuracy of their model was 84% at an individual image level. They also reported that distinguishing the various apical views was the greatest challenge in the setting of partially obscured left ventricles. They made their source code and model weights publicly available at [18]. Mandani et al. [19] presented the classification of 15 standard echo views using DL. They trained a VGG CNN network with 180,294 images of 213 studies and tested their model on 21,747 images of 27 studies. They obtained 91.7% overall accuracy on the test dataset at a single image level and 97.8% overall accuracy when considering the model’s top two guesses. Akkus et al. [20] trained a CNN inception model with residual connections on 5544 images of 140 patients for predicting 24 Doppler image classes and automating Doppler mitral inflow analysis. They obtained overall accuracy of 97% on the test set that included 1737 images of 40 patients.

Abdi et al. [21,22] trained a fully connected CNN with 6196 apical four chamber (A4C) images that were scored between 0 to 5 to assess the A4C quality of echo images. They used three-fold cross validation and reported an error comparable to intra-rater reliability (mean absolute error: 0.71 ± 0.58). Abdi et al. [23] later extended their previous work and trained a CNN regression architecture that includes five regression models with the same weights in the first few layers for assessing the quality of cine loops across five standard view planes (i.e., apical 2, 3, and 4 chamber views and parasternal short axis views at papillary muscle and aortic valve levels). Their dataset included 2435 cine clips, and they achieved an average of 85% accuracy compared to gold standard scores assigned by experienced echo sonographers on 20% of the dataset. Zhang et al. [16,17] calculated the averaged probability score of views classification across all videos in their study to define an image quality score for each view. They assumed that poor quality cine clips tended to have a more ambiguous view assignment, and the view classification probability could be used for quality assessment. Dong et al. [24] presented a generic quality control framework for fetal ultrasound cardiac four chamber planes (CFPs). Their proposed framework consists of three networks that roughly classify four-chamber views from the raw data, determine the gain and zoom of images, and detect the key anatomical structures on a plane. The overall quantitative score of each CFP was achieved based on the output of the three networks. They used five-fold cross validation to assess their model across 2032 CFPs and 5000 non-CFPs and obtained a mean average precision of 93.52%. Labs et al. [25] trained a hybrid model including CNN and LSTM layers to assess the quality of apical four-chamber view images for three proposed attributes (i.e., foreshortening, gain/contrast, and axial target). They split a dataset of 1039 unique apical four-chamber views into 60:20:20% ratio for training, validation, and testing, respectively, and achieved an average accuracy of 86% on the test set.

View identification and quality assessment of cine clips are the most important pieces of a fully automated echo interpretation pipeline. As shown in Table 2, there is an error range of 3–16% in the current studies for both view identification and quality control. The proposed models were generally trained with a dataset from a single or a few vendors or a single center. Apart from the study of Zhang et al. [16,17], none of the studies shared their source code and model weights for comparisons. In some studies, customized CNN models were used, but not enough evidence or comparisons were shown to support that their choices perform better than the state-of-the-art CNN models such as Resnet, Inception, and Densenet.

### 2.2. Image Segmentation and Quantification

Partitioning of an identified view into the region of interests such as left/right ventricle or atrium, ventricular septum, and mitral/tricuspid valves is necessary to quantify certain biomarkers such as ejection fraction, volume changes, and velocity of septal or distal annulus. Several studies have used DL methods to segment left ventricles from apical four and two chamber views.

Zhang et al. [16,17] presented a fully automated echo interpretation pipeline that includes segmentation of cardiac chambers in five common views and quantification of structure and function. They used five-fold cross validation on 791 images that have manual segmentation of left ventricle and reported the intersection over union metric ranging from 0.72 to 0.90 for the performance of their U-Net-based segmentation model. In addition, they produced automated measurements such as LV ejection fraction (LVEF), LV volumes, LV mass, and global longitudinal strain from the resulting segmentations. Compared to manual measurements, median absolute deviation of 9.7% (*n* = 6407 studies) was achieved for LVEF; median absolute deviation of 15–17% was obtained for LV volume and mass measurements; median absolute deviation of 7.5% (*n* = 419) and 9.0% (*n* = 110) was obtained for strain. They concluded that they obtained cardiac structure measurements comparable with values in study reports. Leclerc et al. [13] studied the state-of-art encoder–decoder type DL methods (e.g., U-Net [28]) for segmenting cardiac structures and made a large dataset (500 patients) publicly available with segmentation labels of end diastole and systole frames. The full dataset is available for download at [29]. They showed that their U-Net-based model outperformed the state-of-the-art non-deep-learning methods for measurements of end-diastolic and end-systolic left ventricular volumes and LVEF. They achieved a mean correlation of 0.95 and an absolute mean error of 9.5 mL for LV volumes and a mean correlation coefficient of 0.80 and an absolute mean error of 5.6% for LVEF. Jafari et al. [30] presented a recurrent CNN and optical flow for segmentation of the left ventricle in echo images. Jafari et al. [14] also presented biplane ejection fraction estimation with POCUS using multi-task and learning and adversarial training. The performance of the proposed model for the segmentation of LV was an average Dice score of 0.92 and, for the automated ejection fraction, was shown to be around an absolute error of 6.2%. Chen et al. [31] proposed an encoder–decoder type CNN with multi-view regularization to improve LV segmentation. The method was evaluated on 566 patients and achieved an average Dice score of 0.88. Oktay et al. [32] incorporated anatomical prior knowledge in their CNN model that allows following the global anatomical properties of the underlying anatomy. Ghorbani et al. [33] used a custom CNN model, named EchoNet, to predict left ventricular end systolic and diastolic volumes (R2 = 0.74 and R2 = 0.70), and ejection fraction (R2 = 0.50). Ouyang et al. [15] trained a semantic segmentation model using atrous convolutions on echocardiogram videos. Their model obtained Dice similarity coefficient of 0.92 for left ventricle segmentation of apical four-chamber view and used a spatiotemporal 3D CNN model with residual connections and predicted ejection fraction with mean absolute errors of 4.1 and 6% for internal and external datasets, respectively. Ouyang et al. [15] de-identified 10,030 echocardiogram videos, resized them into 112 × 112 pixels, and made their dataset publicly available at [34].

U-Net is the most common DL model used for echo image segmentation. As shown in Table 3, the error range for LVEF is ranging between 4 and 10%, while it ranges between 10 and 20% for LV and LA volume measurements.

### 2.3. Disease Diagnosis

Several studies have shown that DL models can be used to assess cardiac diseases (see Table 4). Zhang et al. [16,17] presented a fully automated echo interpretation pipeline for disease detection. They trained a VGG [26] network using three random images per video as an input and provided two prediction outputs (i.e., diseased or normal). The ROC curve performance of their model for prediction of hypertrophic cardiomyopathy, cardiac amyloidosis, and pulmonary hypertension were 0.93, 0.87, and 0.85, respectively. Ghorbani et al. [33] trained a customized CNN model that includes inception connections, named EchoNet, on a dataset of more than 1.6 million echocardiogram images from 2850 patients to identify local cardiac structures, estimate cardiac function, and predict systemic risk factors. The proposed CNN model identified the presence of pacemaker leads with AUC = 0.89, enlarged left atrium with AUC = 0.86, and left ventricular hypertrophy with AUC = 0.75. Ouyang et al. [15] trained a custom model that includes spatiotemporal 3D convolutions with a residual connection network together with semantic segmentation of the left ventricle to predict the presence of heart failure with reduced ejection fraction. The output of the spatiotemporal network and semantic segmentation were combined to classify heart failure with reduced ejection fraction. Their model achieved an area under the curve of 0.97 for predicting heart failure with reduced ejection fraction. Omar et al. [35] trained a modified VGG-16 CNN model on a 3D Dobutamine stress echo dataset to detect wall motion abnormalities and compared its performance to hand-crafted approaches: support vector machines (SVM) and random forests (RF). They achieved slightly better accuracy with the CNN model: RF (72.1%), SVM (70.5%), and CNN (75.0%). In another study, Kusunose et al. [36] investigated whether a CNN model could provide improved detection of wall motion abnormalities. They presented that the area under the AUC produced by the deep-learning algorithm was comparable to that produced by the cardiologists and sonographer readers (0.99 vs. 0.98, respectively) and significantly higher than the AUC result of the resident readers (0.99 vs. 0.90, respectively). Narula et al. [37] trained SVM, RF, and artificial neural network (ANN) with hand-crafted echo measurements (i.e., LV wall thickness, end-diastolic volume, end-systolic volume, and ejection fraction, pulsed-wave Doppler-derived transmitral early diastolic velocity (E), the late diastolic atrial contraction wave velocity (A), and the ratio E/A to differentiate hypertrophic cardiomyopathy (HCM) from physiological hypertrophy seen in athletes (ATH). They reported overall sensitivity and specificity of 87 and 82%, respectively.

Unlike other hand-crafted feature-based ML approaches, the DL approaches may extract features from data beyond human perception. DL-based AI approaches have the potential to support accurate diagnosis and discovering crucial features from echo images. In the near future, these tools may aid physicians in diagnosis and decision making and reduce the misdiagnosis rate.

## 3. Discussion and Outlook

Automated image interpretation that mimics human vision with traditional machine learning has existed for a long time. Recent advances in parallel processing with GPUs and deep-learning algorithms, which extract patterns in images with their self-learning ability, have changed the entire automated image interpretation practice with respect to computation speed, generalizability, and transferability of these algorithms. AI-empowered echocardiography has been advancing and moving closer to be used in routine clinical workflow in cardiology due to the increased demand for standardizing acquisition and interpretation of cardiac US images. Even though DL-based methods for echocardiography provide promising results in diagnosis and quantification of diseases, AI-Echo still needs to be validated with larger study populations including multi-center and multi-vendor datasets. High intra-/inter-variability in echocardiography makes standardization of image acquisition and interpretation challenging. However, AI-Echo will provide solutions to mitigate operator-dependent variability and interpretability. AI applications in cardiac US are more challenging than those in cardiac CT and MR imaging modalities due to patient-dependent factors (e.g., obesity, limited acoustic window, artifacts, and signal drops) and natural US speckle noise pattern. These factors that affect US image quality will remain as challenges with cardiac ultrasound.

Applications of DL in echocardiography are rapidly advancing as evidenced by the growing number of studies recently. DL models have enormous representation power and are hungry for large amounts of data in order to obtain generalization ability and stability. Creating databases with large datasets that are curated and have good quality data and labels is the most challenging and time-consuming part of the whole AI model development process. Although it has been shown that AI-echo applications have superb performance compared to classical ML methods, most of the models were trained and evaluated on small datasets. It is important to train AI models on large multi-vendor and multi-center datasets to obtain generalization and validate on large multi-vendor datasets to increase reliability of a proposed model. An alternative way to overcome the limitation of having small training datasets would be augmenting the dataset with realistic transformations (e.g., scaling, horizontal flipping, translations, adding noise, tissue deformation, and adjusting image contrast) that could help improve generalizability of AI models. On the other hand, realistic transformations need to be used to genuinely simulate variations in cardiac ultrasound images, and transformations-applied images should not create artifacts. Alternatively, generative adversarial networks, which include a generator and a discriminator model, are trained until the model generates images that are not separable by the discriminator. This could be used to generate realistic cardiac ultrasound B-mode images of the heart. Introducing such transformations during the training process will make AI models more robust to small perturbations in input data space.

Making predictions and measurements based on only 2D echo images could be considered as a limitation of AI-powered US systems. Two-dimensional cross section images include limited information and do not constitute the complete myocardium. Training AI models on 3D cardiac ultrasound data that include the entire heart or the structure of interest would potentially improve the diagnostic accuracy of an AI model.

It is important to design AI models that are transparent for the prediction of any disease from medical images. The AI models developed for diagnosis of a disease must elucidate the reasons and motivations behind their predictions in order to build trust in them. Comprehension of the inner mechanism of an AI model necessitates interpreting the activity of feature maps in each layer [39,40,41]. However, the extracted features are a combination of sequential layers and become complicated and conceptual with more layers. Therefore, the interpretation of these features become difficult compared to handcrafted imaging features in traditional ML methods. Traditional ML methods are designed for separable components that are more understandable, since each component of ML methods has an explanation but usually is not very accurate or robust. With DL-based AI models, the interpretability is given up for the robustness and complex imaging features with greater generalizability. Recently, a number of methods have been introduced about what DL models see and how to make their predictions. Several CNN architectures [26,28,38,42,43] employed techniques such as deconvolutional networks [44], gradient back-propagation [45], class activation maps (CAM) [41], gradient-weighted CAM [46], and saliency maps [47,48] to make CNN understandable. With these techniques, gradients of a model have been projected back to the input image space, which shows what parts in the input image contribute the most to the prediction outcome that maximizes the classification accuracy. Although making AI models understandable has been an active research topic in the DL community, there is still much further research needed in the area. Despite the fact that high prediction performances were achieved and reported in the studies discussed in this review, none of the studies have provided an insight on which heart regions play an important role in any disease prediction.

Developing AI models that standardize image acquisition and interpretation with less variability is essential considering that echocardiography is an operator- and interpreter-dependent imaging modality. AI guidance during data acquisition for the optimal angle, view, and measurements would make echocardiography less operator-dependent and smarter, while standardizing data acquisition. Cost-effective and easy access of POCUS systems with AI capability would help clinicians and non-experts perform swift initial examinations on patients and progress with vital and urgent decisions in emergency and primary care clinics. In the near future, POCUS systems with AI capability could replace the stethoscopes that doctors use in their daily practice to listen to patients’ hearts. Clinical cardiac ultrasound or POCUS systems empowered with AI, which can assess multi-mode data, steer sonographers during acquisition, and deliver objective qualifications, measurements, and diagnoses, will assist with decision making for diagnosis and treatments, improve echocardiography workflow in clinics, and lower healthcare cost.

## Figures and Tables

**Figure 1 jcm-10-01391-f001:**
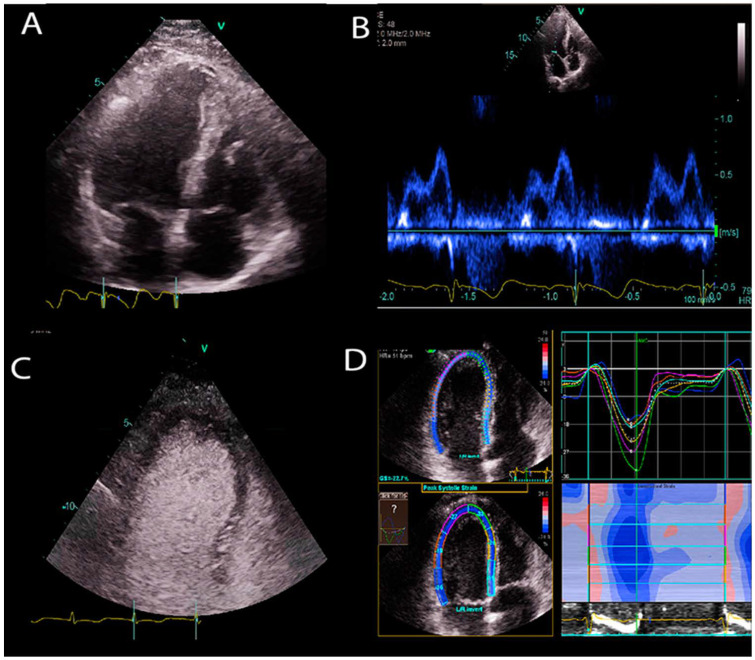
Sample US images showing different US modes. (**A**) B-mode image of the apical 4 chamber view of a heart. (**B**) Doppler image of mitral inflow. (**C**) Contrast enhanced ultrasound image of left ventricle. (**D**) Strain imaging of the left ventricle.

**Figure 2 jcm-10-01391-f002:**
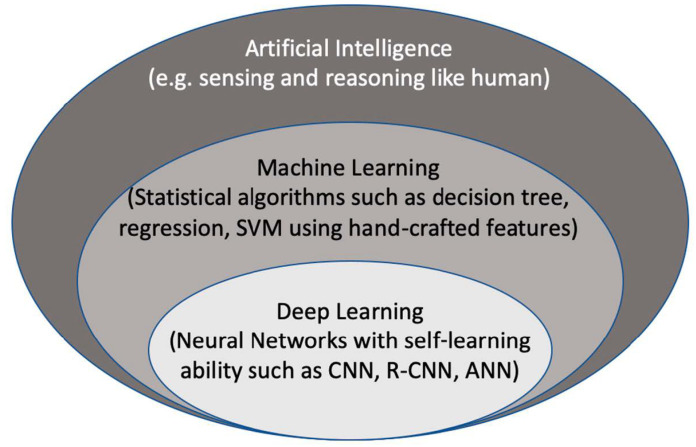
The context of artificial intelligence, machine learning, and deep learning. SVM: Support Vector Machine. CNN: convolutional neural networks, R-CNN: recurrent CNN, ANN: artificial neural networks.

**Figure 3 jcm-10-01391-f003:**
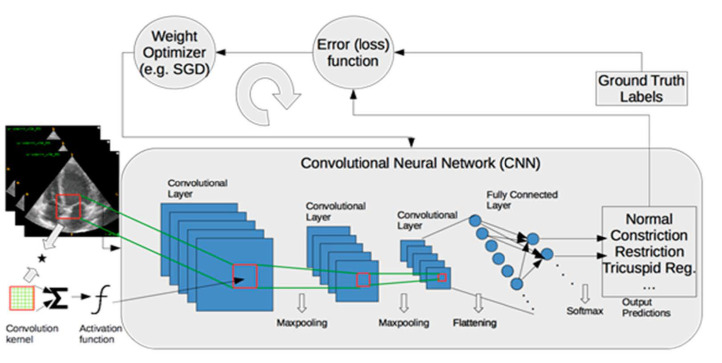
A framework of training a deep-learning model for classification of myocardial diseases. Operations between layers are shown with arrows. SGD: Stochastic Gradient Descent.

**Figure 4 jcm-10-01391-f004:**
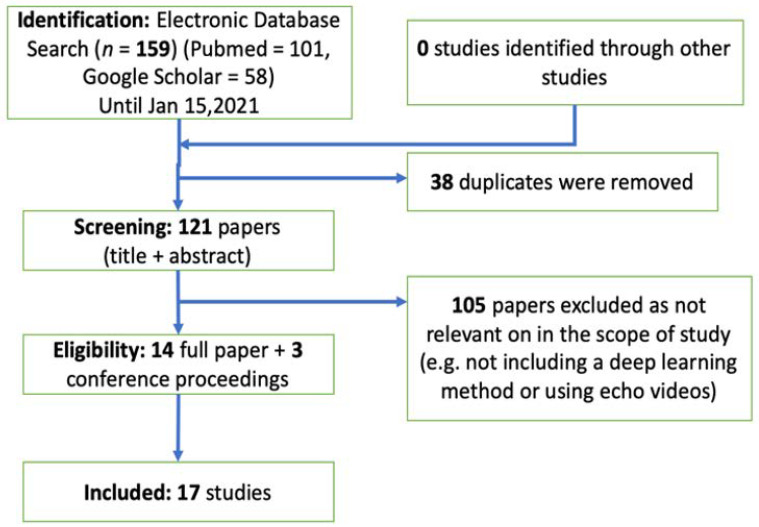
The flowchart of systematic review that includes identification, screening, eligibility, and inclusion.

**Figure 5 jcm-10-01391-f005:**
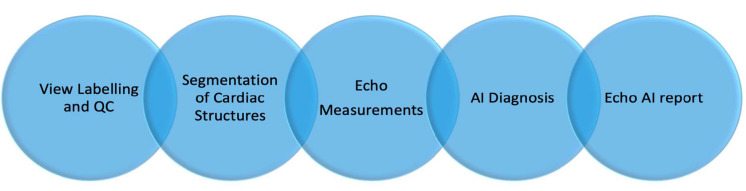
The flowchart of automated artificial-intelligence-empowered echo (AI-Echo) interpretation pipeline using a chain approach. QC: Quality Control.

**Figure 6 jcm-10-01391-f006:**
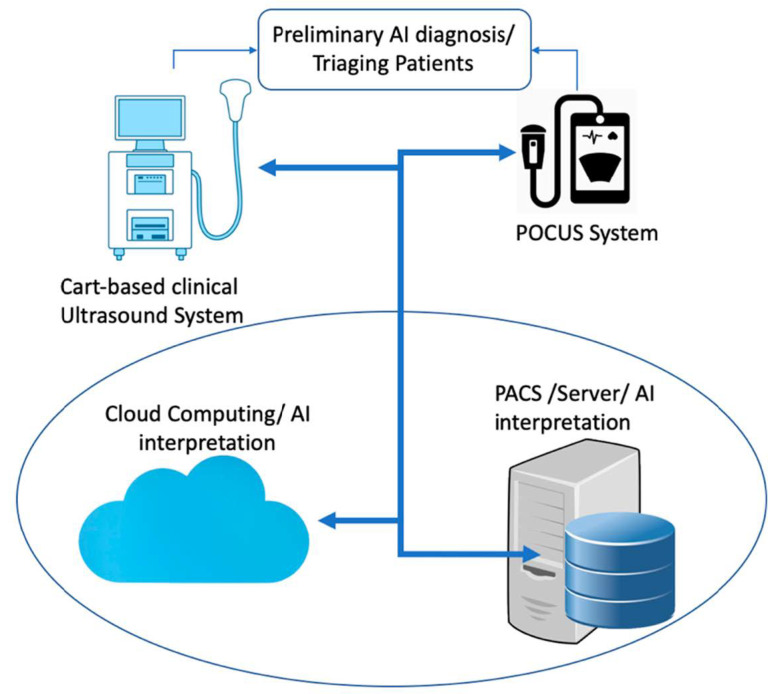
A schematic diagram of AI (artificial intelligence) interpretation of echocardiography images for preliminary diagnosis and triaging patients in emergency and primary care clinics. POCUS: point of care ultrasound.

**Table 1 jcm-10-01391-t001:** The list of commercial software packages that provides automated measurements or diagnosis.

Company	Software Package	AI-Empowered Tools
Siemens Medical Solutions Inc., USA	syngo Auto Left Heart, Acuson S2000 US system.	Auto EF, Auto LV and LA volumes, Auto Strain for manually selected views.
GE Healthcare, Inc., USA	Ultra Edition Package, Vivid Ultrasound Systems	Auto EF, Auto LV and LA volumes, Auto Strain for manually selected views
TOMTEC Imaging Systems GmbH, Germany	Tomtec-Arena/Tomtec-Zero	Auto EF, Auto LV and LA volumes, Auto Strain for manually selected views
Ultromics Ltd., United Kingdom	Echo Go/Echo Go Pro	Auto EF, Auto LV and LA volumes, Auto Strain, Auto identification of CHD (Fully automated)
Dia Imaging Analysis Ltd., Israel	DiaCardio’s LVivoEF Software/LVivo Seamless	Auto EF and Auto standard echo view identification (Fully automated)
Caption Health, Inc., USA	The Caption Guidance software	AI tool for assisting to capture images of a patient’s heart

EF: ejection fraction. CHD: coronary heart disease.

**Table 2 jcm-10-01391-t002:** Deep-learning-based AI studies for view identification and quality assessment. MAE: mean absolute error.

	Task	DL Model	Data/Validation	Performance
Zhang et al. [16,17]	23 standard echo view classification	Customized 13-layer CNN model	5-fold cross validation/7168 cine clips of 277 studies	Overall accuracy: 84% at individual image level
Mandani et al. [19]	15 standard echo view classification	VGG [26]	Training: 180,294 images of 213 studies Testing: 21,747 images of 27 studies	Overall accuracy: 97.8% at individual image level and 91.7% at cine-lip level
Akkus et al. [20]	24 Doppler image classes	Inception_resnet [27]	Training: 5544 images of 140 studies Testing: 1737 images of 40 studies	Overall accuracy of 97%
Abdi et al. [21,22]	Rating quality of apical 4 chamber views (0–5 scores)	A customized fully connected CNN	3-fold cross validation/6196 images	MAE: 0.71 ± 0.58
Abdi et al. [23]	Quality assessment for five standard view planes	CNN regression architecture	Total dataset: 2435 cine clips Training: 80% Testing: 20%	Average of 85% accuracy
Dong et al. [24]	QC for fetal ultrasound cardiac four chamber planes	Ensembled three CNN model	5-fold cross validation (7032 images)	Mean average precision of 93.52%.
Labs et al. [25]	Assessing quality of apical 4 chamber view	Hybrid model including CNN and LSTM layers	Training/validation/testing (60/20/20%) of in total of 1039 images	Average accuracy of 86% on the test set

**Table 3 jcm-10-01391-t003:** Deep-learning-based AI studies for image segmentation and quantification. MAD: mean absolute difference. LVEF: left ventricle ejection fraction.

	Task	DL Model	Data/Validation	Performance
Zhang et al. [16,17]	LV/LA segmentation; LVEF, LV and LA volumes, LV mass, global longitudinal strain	U-Net [28]	LV segmentation: 5-fold cross validation on 791 images; LV volumes: 4748 measurements; LV mass: 4012 measurements; strain: 526 studies	IOU: 0.72–0.90 for LV segmentation; MAD of 9.7% for LVEF; MAD of 15–17% for LV/LA volumes and LV mass; MAD of 9% for strain.
Leclerc et al. [13]	LVEF, LV volumes	U-Net [28]	500 patients	LVEF: AME of 5.6% LV volumes: AME of 9.7 mL
Jafari et al. [14]	LV segmentation and bi-plane LVEF	A shallow U-Net with multi-task learning and adversarial training	854 studies split into 80% training and 20% testing sets	DICE of 0.92 for LV segmentation; MAE of 6.2% for LVEF
Chen et al. [31]	LV segmentation in apical 2, 3, 4, or 5 chamber views	An encoder–decoder type CNN with multi-view regularization	Training set: 33,058 images; test set: 8204 images	Average DICE of 0.88
Oktay et al. [32]	LV segmentation; LVEF	Anatomically constrained CNN model	CETUS’14 3D US challenge dataset. (training set: 15 studies; test set: 30 studies)	DICE of 0.91 ± 0.23 for LV segmentation; correlation of 0.91 for LVEF
Ghorbani et al. [33]	LV systolic and diastolic volumes; LVEF	A customized CNN model (EchoNet) for semantic segmentation	Training set: 1.6 million images from 2850 patients; test set: 169,000 images from 373 studies	Systolic and diastolic volumes (R2 = 0.74 and R2 = 0.70); R2 = 0.50 for LVEF
Ouyang et al. [15]	LVEF	3D CNN model with residual connections	Training set: 7465 echo videos; internal test dataset (*n* = 1277); external test dataset (*n* = 2895)	MAE of 4.1% and 6% for internal and external datasets

**Table 4 jcm-10-01391-t004:** Deep-learning-based AI studies for disease diagnosis. AUC: area under the curve.

	Task	DL Model	Data/Validation	Performance
Zhang et al. [16,17]	Diagnosis of hypertrophic cardiomyopathy (HCM), cardiac amyloidosis (amyloid), and pulmonary hypertension (PAH)	VGG [26]	HCM: 495/2244 Amyloid:179/804 PAH:584/2487 (Diseased/Control) 5-fold cross validation	Hypertrophic cardiomyopathy: AUC of 0.93; cardiac amyloidosis: AUC of 0.87; pulmonary hypertension: AUC of 0.85
Ghorbani et al. [33]	Diagnose presence of pacemaker leads; enlarged left atrium; LV hypertrophy	A customized CNN model	Training set: 1.6 million images from 2850 patients; test set: 169,000 images from 373 studies	Presence of pacemaker leads with AUC = 0.89; enlarged left atrium with AUC = 0.86, left ventricular hypertrophy with AUC = 0.75.
Ouyang et al. [15]	Predict presence of HF with reduced EF	3D convolutions with residual connection	Training set: 7465 echo videos; internal test dataset (*n* = 1277); external test dataset (*n* = 2895)	AUC of 0.97
Omar et al. [35]	Detecting wall motion abnormalities	Modified VGG-16 [26]	120 echo studies. One-leave-out cross validation	Accuracy: RF = 72.1%, SVM = 70.5% CNN = 75.0%
Kusunose et al. [36]	Detecting wall motion abnormalities (WMA)	Resnet [38]	300 patients with WMA +100 normal control. Training = 64% Validation:16% Test: 20%	AUC of 0.99
Narula et al. [37]	Differentiate HCM from ATH	A customized ANN	77 ATH and 62 HCM patients. Ten-fold cross validation	Sensitivity: 87% Specificity: 82%

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
