# Peer review of "Artificial Intelligence (AI)-Empowered Echocardiography Interpretation: A State-of-the-Art Review"

_jcm, 2021, doi:10.3390/jcm10071391_

Round 1

Reviewer 1 Report

The authors present a comprehensive review of the existing research using deep learning techniques in echocardiography. These investigators have provided a nice commentary on the current state of machine learning / deep learning in echocardiography, what has been accomplished, and the challenges anticipated with developing robust deep learning models. The comment about the need for large, shared datasets is particularly insightful and forward-thinking. I have only a couple thoughts. 

Figure 2 - I'm not sure this is the correct way to present artificial intelligence, machine learning, and deep learning. While these techniques are all interrelated, I'm not sure they are nested in the way the authors present. Or if the authors have good evidence to support why they are nested they should explain this relationship more clearly in the text; or at least in the figure legend. 

The discussion section commentary is nice. However, I wonder if the authors - having a very good handle on the existing literature - could make a comment on future research on deep learning in echocardiography. Where are the gaps, and where are the opportunities?

Author Response

Dear Editor,

We would like to thank the reviewer for the constructive comments and we provided the response for each comment below.

Reviewer 1:

The authors present a comprehensive review of the existing research using deep learning techniques in echocardiography. These investigators have provided a nice commentary on the current state of machine learning / deep learning in echocardiography, what has been accomplished, and the challenges anticipated with developing robust deep learning models. The comment about the need for large, shared datasets is particularly insightful and forward-thinking. I have only a couple thoughts. 

Figure 2 - I'm not sure this is the correct way to present artificial intelligence, machine learning, and deep learning. While these techniques are all interrelated, I'm not sure they are nested in the way the authors present. Or if the authors have good evidence to support why they are nested they should explain this relationship more clearly in the text; or at least in the figure legend. 

Answer:  In page 3, first paragraph, we describe the relationship between artificial intelligence, machine learning, and deep learning. In the literature, the relationship between them is shown in the same way at Figure 2. Deep learning is considered as subclass of Machine Learning (ML) with its self-learning ability of feature extraction and AI encompasses both ML and AI.

“Artificial intelligence (AI) is considered to be a computer-based system that can observe an environment and takes actions to maximize the success of achieving its goals. Some examples include a system that has the ability of sensing, reasoning, engaging, and learning, are computer vision for understanding digital images, natural language processing for interaction between human and computer languages, voice recognition for detection and translation of spoken languages, robotics and motion, planning and organization, and knowledge capture. ML is a subsection of AI that covers the ability of a system to learn about data using supervised or unsupervised statistical and ML methods such as regression, support vector machines, decision trees, and neural networks. Deep learning (DL), which is a subclass of ML, learns a sequential chain of pivotal features from input data that maximizes the success of the learning process with its self-learning ability. This is different from statistical ML algorithms that require handcrafted feature selection (3) (Figure 2).”

The discussion section commentary is nice. However, I wonder if the authors - having a very good handle on the existing literature - could make a comment on future research on deep learning in echocardiography. Where are the gaps, and where are the opportunities?

Answer: As shown the in the page 14, lines 446-459, we explained the future direction of echo AI. There is still gap in optimal data acquisition and integrating POCUS into the clinical workflow for preliminary diagnosis using AI as mentioned below.

“Developing AI models that standardize image acquisition and interpretation with less variability is essential considering that echocardiography is an operator and interpreter dependent imaging modality. AI guidance during data acquisition for the optimal angle, view, and measurements would make echocardiography less operator dependent and smarter, while standardizing data acquisition. Cost-effective and easy access of POCUS systems with AI capability would help clinicians and non-experts perform swift initial examinations on patients and progress with vital and urgent decisions in emergency and primary care clinics. In the near future, POCUS systems with AI capability could replace the stethoscopes that doctors use in their daily practice to listen to patients' hearts. Clinical cardiac ultrasound or POCUS systems empowered with AI, which can assess multi-mode data, steer sonographers during acquisition, and deliver objective qualifications, measurements and diagnosis, will assist with decision making for diagnosis and treatments, improve echocardiography workflow in clinics, and lower healthcare cost.”

Sincerely,

Zeynettin Akkus

Reviewer 2 Report

The paper presents the role of artificial intelligence in echocardiography field. Overall, the scientific objective is important and interesting. The article is well written and comprehensive. There are some minor revisions needed. Please provide a point-by-point response to the following queries.

  1. Please explain the abbreviations used in the figures and tables in their legends.
  2. Was the review done according to PRISMA Guidelines?
  3. The methodology section should provide information on full electronic search, including any limits used, such that it could be repeated. Specify study characteristics and report characteristics (e.g., years considered, language, publication status) used as criteria for eligibility, giving rationale.
  4. Please provide a flowchart, which depicts the flow of information through the different phases of a systematic review. It should include detailed information on the number of records identified, included and excluded, and the reasons for exclusions.
  5. The results section should include information on numbers of studies screened, assessed for eligibility, and included in the review, with reasons for exclusions at each stage.

Author Response

Dear Editor,

We would like to thank the reviewer for the constructive comments and we provided the response for each comment below. All changes highlighted with yellow background in the revised manuscript.

Reviewer 2:

The paper presents the role of artificial intelligence in echocardiography field. Overall, the scientific objective is important and interesting. The article is well written and comprehensive. There are some minor revisions needed. Please provide a point-by-point response to the following queries.

  1. Please explain the abbreviations used in the figures and tables in their legends.

Answer: Thanks to the reviewer. We explained abbreviations in their legends.

  1.  Was the review done according to PRISMA Guidelines?

Answer:  Yes, we followed the PRISMA guidelines to structure our review. We thought merging method and results section as “automated echo interpretation” would be more appropriate for this review.

  1.  The methodology section should provide information on full electronic search, including any limits used, such that it could be repeated. Specify study characteristics and report characteristics (e.g., years considered, language, publication status) used as criteria for eligibility, giving rationale.

Answer: Thanks to the reviewer. We added this to the methodology section.

  1. Please provide a flowchart, which depicts the flow of information through the different phases of a systematic review. It should include detailed information on the number of records identified, included and excluded, and the reasons for exclusions.

            Answer: We included the flowchart for paper identification, selection, inclusion, and exclusion.    (Please see Figure 4 in the revised manuscript)

  1.  The results section should include information on numbers of studies screened, assessed for eligibility, and included in the review, with reasons for exclusions at each stage.

Answer: Thank to the reviewer. The number of studies screened, included and excluded in the review are shown in the Figure 4. Also, we included in the text.

Sincerely,

Zeynettin Akkus